# The Roles of Caloric Restriction Mimetics in Central Nervous System Demyelination and Remyelination

**Despoina Kaffe [1], Stefanos Ioannis Kaplanis [2,3] and Domna Karagogeos [2,3,\*]**

[1]   Department of Biology, University of Crete, Vassilika Vouton, 70013 Heraklion, Greece;
    bio1p443@edu.biology.uoc.gr
[2]   Department of Basic Science, School of Medicine, University of Crete, Vassilika Vouton,
    70013 Heraklion, Greece; ioannis_kaplanis@imbb.forth.gr
[3]   Institute of Molecular Biology & Biotechnology (IMBB), Foundation for Research and Technology-Hellas
    (FORTH), Vassilika Vouton, 70013 Heraklion, Greece
\*   Correspondence: karagoge@imbb.forth.gr; Tel.: +30-2810394542

**Abstract:** The dysfunction of myelinating glial cells, the oligodendrocytes, within the central nervous system (CNS) can result in the disruption of myelin, the lipid-rich multi-layered membrane structure that surrounds most vertebrate axons. This leads to axonal degeneration and motor/cognitive impairments. In response to demyelination in the CNS, the formation of new myelin sheaths occurs through the homeostatic process of remyelination, facilitated by the differentiation of newly formed oligodendrocytes. Apart from oligodendrocytes, the two other main glial cell types of the CNS, microglia and astrocytes, play a pivotal role in remyelination. Following a demyelination insult, microglia can phagocytose myelin debris, thus permitting remyelination, while the developing neuroinflammation in the demyelinated region triggers the activation of astrocytes. Modulating the profile of glial cells can enhance the likelihood of successful remyelination. In this context, recent studies have implicated autophagy as a pivotal pathway in glial cells, playing a significant role in both their maturation and the maintenance of myelin. In this Review, we examine the role of substances capable of modulating the autophagic machinery within the myelinating glial cells of the CNS. Such substances, called caloric restriction mimetics, have been shown to decelerate the aging process by mitigating age-related ailments, with their mechanisms of action intricately linked to the induction of autophagic processes.

**Keywords:** oligodendrocyte progenitor cells; oligodendrocytes; microglia; astrocytes; caloric restriction; caloric restriction mimetics; demyelination; remyelination; autophagy

## 1. Introduction

Myelin is a lipid-rich multi-layered membrane structure that surrounds most vertebrate axons. It is characterized by a high lipid-to-protein ratio, containing 75–80% lipids (by dry weight) and 25–30% proteins [1]. A hallmark of myelinated fibers is saltatory conduction, a mechanism that enables the rapid and efficient propagation of action potentials along the axonal length. Beyond its insulating function, myelin also plays an active role in providing metabolic support to axons [2]. In the central nervous system (CNS), myelin is produced by oligodendrocytes (OLs), while in the peripheral nervous system (PNS), it is synthesized by Schwann cells (SCs) [3].

The dysfunction of myelinating glial cells within the CNS can result in the disruption of myelin, a phenomenon that can subsequently lead to axonal demyelination and contribute to eventual axonal degeneration [4]. In response to demyelination in the CNS, the formation of new myelin sheaths occurs through the homeostatic process of remyelination, facilitated by the differentiation of newly formed OLs [5]. It is important to note, however, that remyelination is often inadequate for fully replicating the original myelin ultrastructure [5].

Apart from OLs, microglia and astrocytes, the two other main glial cell types of the CNS, play a pivotal role in remyelination [6–8]. Microglia constitute the resident macrophages within the CNS and display dynamic diversity [9,10]. Following a demyelination insult, they can phagocytose myelin debris, which is vital for recruiting and differentiating oligodendrocyte progenitor cells (OPCs), and they secrete growth factors and chemotactic substances. They also alter the extracellular matrix to support OPCs, aiding remyelination [11–14]. The developing neuroinflammation in the demyelinated region triggers the activation of astrocytes, the third glial population involved in remyelination, in a process known as reactive astrogliosis [15]. Activated astrocytes play a dual role, sometimes favoring or hindering remyelination based on their specific phenotype [16,17]. Astrocytes can directly impact remyelination because they recruit microglia to the lesion site and, thus, modulate the removal of myelin debris, which is essential for the resolution of the inflammatory response and, ultimately, remyelination [18]. Additionally, they modulate the extracellular matrix, affecting OPC proliferation and differentiation [19]. Modulating the expression profile of these glial cell types can enhance the likelihood of successful CNS remyelination.

In addition, recently, studies have implicated autophagy as a pivotal pathway in glial cells, playing a significant role in both their maturation and the maintenance of myelin. A study by Bankston and colleagues reveals that autophagy deficiency hinders OL differentiation both in vitro and in vivo. Specifically, their findings indicate that inhibiting autophagy alters the ultrastructure of myelin, thus underscoring the critical role of autophagy in OLs for proper myelination. Additionally, the observed enrichment of autolysosomes in OL processes suggests a specialized role of autophagy in these cellular processes. Notably, this research group also reports an increase in autophagic flux during oligodendrocyte differentiation in vitro [20]. Two recent consecutive studies have highlighted the essential role of autophagy in OLs not only in myelination but also in maintaining myelin throughout the lifespan of mice. Aber et al. demonstrate that OLs orchestrate the autophagic machinery to turnover myelin sheaths during adulthood, as autophagy deficiency leads to increased myelin deposition, a phenomenon that intensifies over time [21]. Furthermore, Ktena et al. corroborate the hypothesis that autophagy plays a pivotal role in myelin maintenance. The inhibition of autophagy, both genetically and pharmacologically, results in defects in OL maturation in vitro. The ablation of the core autophagic gene atg5 in OLs in vivo in 2.5 month-old mice, following the completion of myelination, leads to an excess of both PLP protein and mRNA levels at the age of 6 months, also implicating the autophagic machinery in PLP mRNA degradation. Moreover, conditional knockout mice in which autophagy is ablated in OLs exhibit myelin decompaction with subsequent axonal degeneration and behavioral deficits at the age of 6 months [22]. Collectively, these recent findings regarding autophagy and its pivotal role in CNS myelination and myelin maintenance have made autophagy an attractive therapeutic target for repairing myelin insults and/or abnormalities.

In the context of this Review, we focused on substances capable of modulating the autophagic machinery within the myelinating glial cells of the CNS. This modulation is achieved through a process akin to caloric restriction (CR), and the substances are named caloric restriction mimetics (CRMs) [23,24]. Dietary or caloric restriction is defined as the deliberate reduction of food consumption while maintaining proper nutrition, irrespective of the selective restriction of specific food groups. After nearly a century of extensive investigation across various model organisms, including *Saccharomyces cerevisiae*, *Drosophila melanogaster*, rodents, and non-human primates, and the analysis of human epidemiological data, CR is presently widely acknowledged for its capacity to enhance the longevity of organisms and decelerate the aging process [25,26]. Furthermore, it is recognized for its ability to mitigate age-related ailments, with its mechanisms of action being intricately linked to the induction of autophagic processes [27].

In the specific context of the CNS myelin, the advantageous effects of CR have been demonstrated. Piccio and colleagues provided compelling evidence illustrating the efficacy

of a chronic CR regimen in enhancing the clinical outcomes of both relapsing–remitting and chronic experimental autoimmune encephalomyelitis (EAE) models [23]. These improvements in clinical outcomes were further validated through the observation of reduced severity in CNS pathology among the mice subjected to CR. Furthermore, the beneficial impact of CR on myelin recovery has also been observed in the cuprizone (CPZ) model of demyelination. Studies have shown that CR fosters the remyelination process by significantly increasing the survival rates of OLs. Additionally, it leads to a decrease in both astrogliosis and microgliosis within the corpus callosum (cc) of mice with CPZ-induced demyelination [24]. Nevertheless, owing to the systemic and extensive impacts of CR, unraveling the specific signaling pathways and the exact mechanisms underpinning its favorable effects mediated via autophagy can prove to be a complex endeavor. Over the course of decades of research, several hypotheses have arisen, among which the predominant one suggests that CR primarily acts to preserve cellular homeostasis and overall health [28].

The depletion of nutrients leads to a reduction in intracellular acetyl coenzyme A (AcCoA) levels, concurrent with the deacetylation of cellular proteins. Within this conceptual framework, there are three potential approaches to replicate these effects: (i) decrease cytosolic AcCoA levels by disrupting its biosynthesis; (ii) inhibit acetyltransferases, enzymes responsible for transferring acetyl groups from AcCoA to various molecules; or (iii) promote the activity of deacetylases, which facilitate the removal of acetyl groups from leucine residues [27,29,30]. The impact of CR can be replicated through the use of specific pharmacological agents referred to as CRMs. These agents, including metformin, nicotinamide adenine dinucleotide (NAD+) precursors, and resveratrol, are non-toxic natural compounds, which exhibit the capability to modulate the autophagic flux by triggering pathways similar to those activated during nutrient deprivation [31]. For the reasons mentioned above, we will focus this Review on the roles of metformin, NAD+ precursors, and resveratrol, mainly in demyelinating diseases.

## 2. Metformin

Metformin is a derivative of the natural guanidines present in the plant *Galega officinalis* and is widely used as a drug for type II diabetes, primarily operating through the inhibition of hepatic gluconeogenesis [32–34]. Beyond its established role in managing type II diabetes, metformin administration seems to exert beneficial effects on diseases, including cancer [35,36], cardiovascular disease [37], and obesity [38], as well as on neurodegeneration [39] and aging [40]. However, the precise underlying mechanisms responsible for these diverse therapeutic benefits remain to be elucidated [41].

Metformin inhibits mitochondrial complex I [42], a crucial component of the electron transport chain, thereby leading to decreased cellular ATP/ADP and ATP/AMP ratios and, thus, adenosine 5′-monophosphate-activated protein kinase (AMPK) activation [43,44]. Importantly, metformin-mediated AMPK activation exerts regulatory effects on cell energy metabolism and the autophagic cascade by reducing the activity of EP300 acetyl-transferase [45] and simultaneously enhancing the activity of sirtuin 1 (SIRT1) protein deacetylase [46]. Furthermore, a recent study has highlighted an additional mode of AMPK activation by metformin, which directly acts on the lysosomal vacuolar-type ATPase (v-ATPase), promoting the formation of the v-ATPase-regulator-AXIN/liver kinase B1 (LKB1)-AMPK complex in the lysosome, ultimately leading to AMPK activation. Interestingly, when the v-ATPase-regulator complex is engaged by AXIN, it inactivates mammalian target of rapamycin complex 1 (mTORC1), demonstrating that metformin's effects extend beyond AMPK activation, also encompassing mTORC1 inactivation [47]. The activation of AMPK together with the inactivation of mTORC1, the two major energy and nutrient sensors of the cell, induce the activation of the autophagic pathway [48].

It has long been reported that metformin exerts neuroprotective effects on several neurodegenerative diseases, such as Alzheimer's disease (AD), Parkinson's disease (PD), and Huntington's disease (HD). Chronic metformin administration was found to amelio-

rate synaptic malfunctions and cognitive impairment in the amyloid precursor protein (APP)swe/presenilin-1(PS1)DE9 (APP/PS1) mouse model of AD via the inhibition of cyclin-dependent kinase 5 (Cdk5) activity [49]. In the same mouse model of early-onset AD, metformin promoted the phagocytosis of pathological amyloid-β (Aβ) and tau proteins by microglia via the enhancement of the autophagic pathway, thus reducing the abundance of Aβ deposits and severity of neuritic plaque (NP) tau-pathology [50]. Metformin was also found to exert neuroprotective effects on dopaminergic neurodegeneration and alpha-synuclein aggregation in *Caenorhabditis elegans* models of PD [51], while it alleviated motor and neuropsychiatric manifestations in the zQ175 mouse model of HD [52].

Even though most of the current evidence suggests a beneficial effect of metformin on the prevention of AD in humans, its efficacy seems to be controversial. Recently, an observational study has indicated that metformin was associated with slower cognitive decline and reduced risk of dementia in patients with type II diabetes [53]. Furthermore, a randomized, double-blinded, placebo-controlled crossover pilot study demonstrated that metformin is safe and well tolerated by individuals, while being able to penetrate the blood–brain barrier [54]. Interestingly, metformin improved the executive function and tended to ameliorate memory, learning, and attention [54]. However, results from a prospective trial revealed that metformin impaired cognitive performance and that this effect was, at least in part, mediated by metformin-induced vitamin B12 deficiency [55]. This controversy could be attributed to different sample sizes, statistical methods, and drug administration, suggesting that more clinical trials need to be conducted [40].

In line with the multitude of evidence indicating metformin's favorable results in neurodegenerative conditions, many recent studies have diligently scrutinized the potential therapeutic implications of metformin in the context of multiple sclerosis (MS) [56]. MS, a complex and heterogeneous neurodegenerative disorder affecting the CNS, is primarily characterized by profound demyelination, inflammation, and reactive gliosis [57]. Metformin treatment was shown to protect against intense demyelination in the cc of the CPZ-induced demyelination mouse model, when administered with the copper chelator CPZ, by attenuating reactive microgliosis and astrogliosis in the cc (Figures 1 and 2) [58,59].

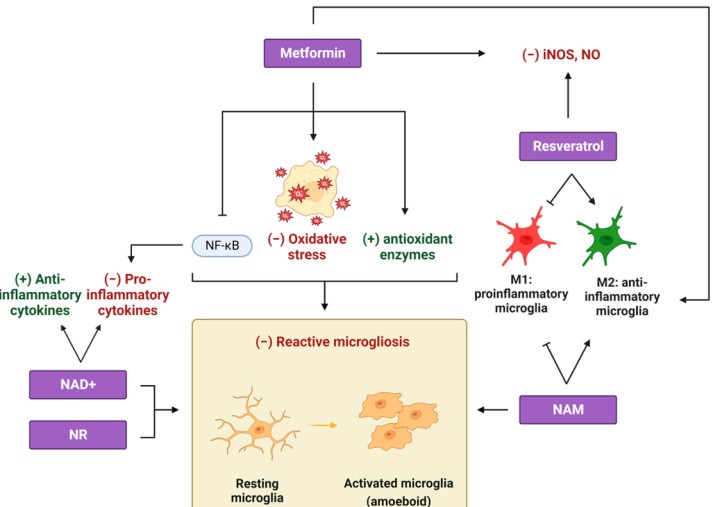

**Figure 1.** The effects of different caloric restriction mimetics (CRMs) on microglia. Metformin treatment reduces oxidative stress, upregulates antioxidant enzymes, and downregulates NF-κB signaling in microglia, thus attenuating the production of pro-inflammatory cytokines, ultimately leading to reduced microgliosis. Moreover, a similar reduction in reactive microgliosis is observed following treatment with NR, NAD+, and NAM. NAD+ administration results in a decreased expression of pro-inflammatory cytokines and an increased expression of anti-inflammatory ones. In parallel, NAM facilitates the polarization of microglia toward their anti-inflammatory phenotype, an effect that is also evident in response to metformin and resveratrol treatment, which additionally re-

duce iNOS and NO levels. The cumulative impact of these cellular responses contributes to the establishment of a less inflammatory milieu that could support the remyelination process. iNOS: inducible nitric oxide synthase; NF-κB: nuclear factor kappa-light-chain-enhancer of activated B cells; NR: nicotinamide riboside; NAD+: nicotinamide adenine dinucleotide; NAM: nicotinamide; NO: nitric oxide. Created with BioRender.com (accessed on 21 November 2023).

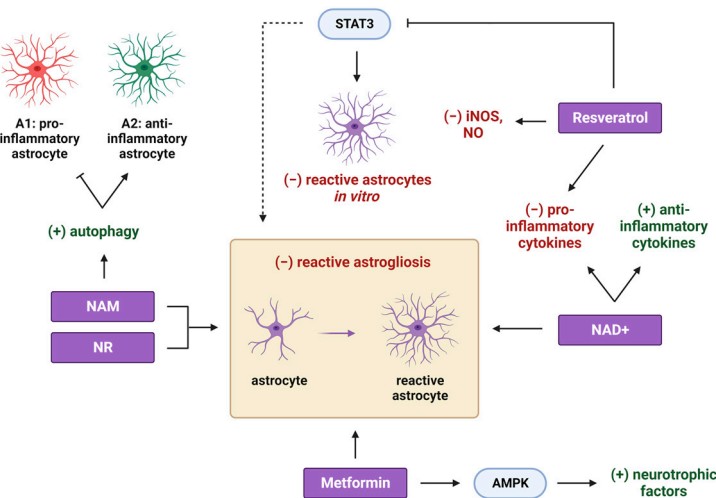

**Figure 2.** The effects of different CRMs on astrocytes. Resveratrol attenuates the expression of pro-inflammatory cytokines and inflammatory markers, such as iNOS and NO, while it also down-regulates STAT3 signaling, leading to reduced proliferation and activation of reactive astrocytes. Given that STAT3 is implicated in the formation of glial scars in response to CNS demyelination, it is possible that resveratrol can reduce reactive astrogliosis in vivo. Furthermore, reduced reactive astrogliosis is observed upon NR, NAD+, NAM, and metformin treatments. NAD+ administration results in a decreased expression of pro-inflammatory cytokines and an increased expression of anti-inflammatory ones. In a similar way, NAM facilitates the polarization of astrocytes toward their anti-inflammatory phenotype, partially through the induction of autophagy. Metformin, in turn, acts through the activation of AMPK and induces the production of neurotrophic factors by astrocytes. This protective, less inflammatory microenvironment can support the process of remyelination in vivo after a demyelination insult. STAT3: signal transducer and activator of transcription 3; AMPK: adenosine 5′-monophosphate-activated protein kinase. Created with BioRender.com (accessed on 21 November 2023).

Largani et al. attributed the beneficial effects of metformin on myelin maintenance and reduced gliosis to its ability to reduce oxidative stress and upregulate antioxidant enzymes. There have been reports suggesting that reactive oxygen species (ROS) can regulate the expression of pro-inflammatory genes in microglia [60] and stimulate astrocytes to secrete inflammatory cytokines [61]. Abdi and his colleagues showed that metformin reduced levels of pro-inflammatory microglia markers through suppressing nuclear factor kappa-light-chain-enhancer of activated B cells (NF-κB) in the CPZ model of MS, an effect that was accompanied by the delayed initiation of gliosis. Moreover, metformin administration was shown to decrease inducible nitric oxide synthase (iNOS) mRNA levels in EAE mice (Figure 1) [62] as well as to protect myelin and promote an anti-inflammatory microglial phenotype that promotes the clearance of myelin debris in a rat spinal cord injury model (Figure 1) [63]. In this case, the effects of metformin were mediated by the induction of autophagy through the activation of AMPK and the inhibition of mTORC1 [63]. These results indicate that metformin can act in favor of a less inflammatory environment under demyelinating conditions, thus enabling the physiological process of remyelination to take over demyelination. In particular, metformin administration during the recovery period significantly promoted the recruitment of intermediate and premature OPCs to the lesion site in favor of the remyelination process in a CPZ-induced demyelination

mouse model [64]. In this case, accelerated myelin recovery upon metformin treatment was mediated by AMPK activation and m-TORC inactivation in mature OLs (Figure 3), indicating a possible implication of the autophagic machinery in the recovery process.

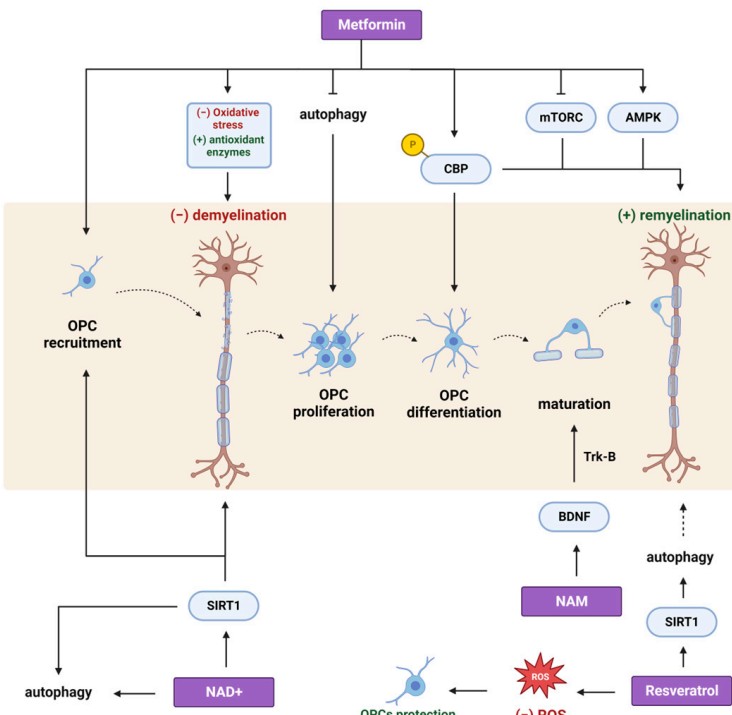

**Figure 3.** The effects of different CRMs on oligodendrocyte lineage cells and the process of re-myelination. Metformin administration promotes the recruitment of OPCs to the lesion site and attenuates demyelination through its ability to reduce oxidative stress and upregulate antioxidant enzymes. Furthermore, metformin promotes OPC proliferation via the blockage of autophagy and enhances OPC differentiation and maturation into myelinating OLs via CBP phosphorylation. The remyelination process is also enhanced by the modulation of AMPK/mTORC pathways in mature OLs. Apart from metformin, NAM and resveratrol can promote the remyelination process. NAM treatment promotes the maturation of OPCs via the BDNF/TrkB pathway. Resveratrol protects OLs by reducing the abundance of ROS while it also induces autophagy through the activation of SIRT1, a mechanism that likely supports remyelination. Finally, direct NAD+ supplementation facilitates the recruitment of OPCs and alleviates demyelination through the activation of the SIRT1 signaling pathway and, probably, through the induction of autophagy. OPCs: oligodendrocyte progenitor cells; OLs: oligodendrocytes; CBP: CREB-binding protein; mTORC: mammalian target of rapamycin complex; BDNF: brain-derived neurotrophic factor; TrkB: tropomyosin receptor kinase B; SIRT1: sirtuin 1. Created with BioRender.com (accessed on 21 November 2023).

Despite the positive impacts of metformin on myelin protection and recovery through the inhibition of mTOR, it has been demonstrated that mTOR signaling regulates the developmental myelination of the CNS [65]. In particular, the ablation of raptor, the defining subunit of mTORC1, in OLs results in impaired OPC differentiation and delayed initiation of myelination in the spinal cords of mutant mice, which are also characterized by the formation of thinner myelin sheaths [65]. Interestingly, mTORC1 signaling driven by phosphoinositide 3-kinase (PI3K)/Akt, rather than ERK1/2, regulates the differentiation of progenitors, whereas both pathways converge at the level of mTORC1 to modulate myelin growth during active myelination [66]. Nevertheless, it has been shown that the prevention of the expression of tuberous sclerosis complex 1 (TSC1), a suppressor of mTOR signaling, resulted unexpectedly in hypomyelination during development [67]. On the other hand, the loss of TSC1 in adult OPCs enhanced remyelination and increased myelin thickness following lysolecithin (LPC)-induced focal demyelination [68], indicating that

the deficiency in mTOR suppressors may exert either beneficial or detrimental effects on the differentially regulated processes of developmental myelination and remyelination. Therefore, more research is warranted to elucidate the interactions between AMPK and mTOR upon metformin administration.

Regarding AMPK, it is a cellular energy regulator found in many types of brain cells, including neurons, astrocytes, and microglia, as well as OLs [69]. Metformin treatment accelerates the differentiation of OLs in an AMPK-dependent manner, also requiring active glycolysis and/or oxidative phosphorylation to mediate OL differentiation [70]. The same study demonstrates the potential of metformin to improve myelin recovery from CPZ-induced demyelination by promoting OL differentiation in vivo [70]. Metformin-mediated AMPK activation seems to protect OLs against cytokine toxicity and oxidative stress rescuing their loss in the spinal cords of EAE rats, thus attenuating the clinical impairments of the disease and restoring the CNS integrity [71]. These immunomodulatory activities of AMPK signaling are concomitant with the stimulation of neurotrophic factor production in astrocytes within the CNS, which subsequently provides a myelinogenic environment for OLs (Figure 1) [71]. Furthermore, metformin treatment leads to increased synthesis of neurotrophic factors, like nerve growth factor (NGF), brain-derived neurotrophic factor (BDNF), and ciliary neurotrophic factor (CNTF), while it also induces the expression of mature oligodendrocyte markers and the activation of AMPK in the CPZ-induced demyelination mouse model [72]. Because neurotrophic factors are reported to enhance OPC survival, migration, proliferation, differentiation, and maturation [73], it is suggested that metformin enhances the secretion of these factors during the recovery phase after demyelination, thus affecting the migration and differentiation of oligodendrocyte transcription factor 2 (Olig2)+ cells in favor of remyelination, effects that are mediated by AMPK activation [72].

Apart from its AMPK-mediated effects on remyelination, metformin was also found to act via the phosphorylation of the histone acetyltransferase CREB-binding protein (CBP), ultimately promoting OPC recruitment and differentiation to the lesion site in an LPC-induced focal demyelination mouse model. In particular, in vitro experiments confirmed that CBP Ser436 phosphorylation is required for metformin to promote the differentiation of OPCs into mature OLs. However, it is not responsible for metformin-induced OPC proliferation, an effect that was connected with the ability of metformin to block autophagy at early stages (Figure 3) [74].

When the remyelination process is delayed or fails, demyelinated axons are susceptible to irreversible degeneration, which can eventually lead to neuronal death [75]. The deceleration of remyelination that occurs in aging is marked by the deficient recruitment of OPCs to the site of the injury, coupled with the delayed progression in their differentiation into mature OLs [76]. Reversing the age-related intrinsic deficiencies of OPCs that are associated with their inability to respond to pro-differentiation factors was able to enhance OPC differentiation and remyelination in aged animals [77]. In this study, metformin was found to improve the mitochondrial function of aged OPCs by modulating the AMPK pathway and to restore the CNS remyelination capacity in aged rats, following ethidium bromide-induced focal demyelination in the cerebellar white matter (Figure 4). The authors postulated that metformin's impact on remyelination could also be attributed to its capacity to enhance DNA repair and induce autophagy, both of which are established effects associated with metformin [78,79].

Regarding its functional behavioral effects, metformin treatment improved motor impairment and reduced anxiety in the CPZ-induced demyelination mouse model [64], while it also improved the social interaction of juvenile mice in an LPC-induced focal demyelination model [74]. These results render metformin a promising remyelinating agent that could treat neural deficits and impaired social behavior, a common symptom of white-matter-demyelinating diseases.

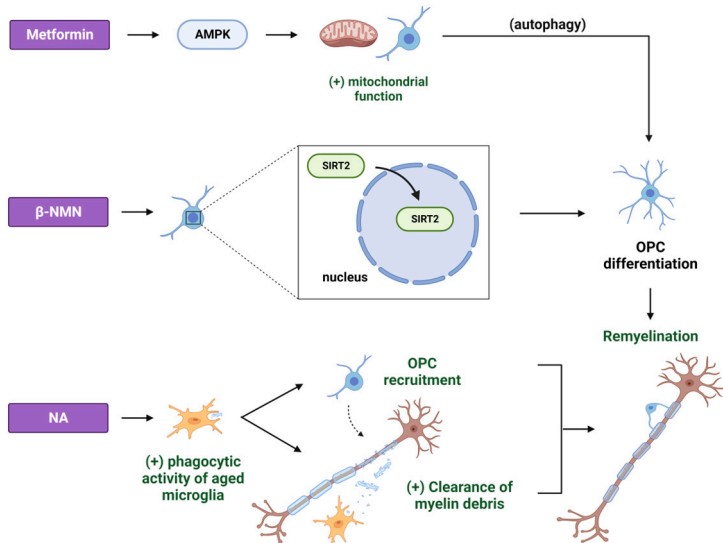

**Figure 4.** The effects of different CRMs on the aged central nervous system (CNS) after a demyelination insult. The administration of metformin improves the mitochondrial function of aged OPCs via the activation of AMPK, consequently facilitating the restoration of the CNS myelination, an effect that could be attributed to the induction of autophagy. Furthermore, β-NMN supplementation rescues SIRT2 nuclear localization in the OPCs of aged mice, thus enhancing remyelination by promoting the differentiation of aged OPCs. Finally, NA promotes remyelination by enhancing the phagocyting activity of middle-aged microglia. This process results in the removal of myelin debris from the lesion site and the recruitment and maturation of OPCs into myelinating OLs. β-NMN: β-nicotinamide mononucleotide; SIRT2: sirtuin 2; NA: nicotinic acid. Created with BioRender.com (accessed on 21 November 2023).

## 3. NAD+ Precursors

Nicotinic acid (NA), commonly referred to as niacin or vitamin B3, along with nicotinamide (NAM), its amide derivative, nicotinamide riboside (NR), and nicotinamide mononucleotide (NMN), serve as precursors for NAD+ (Figure 5). These compounds are available in various dietary products of both animal and plant origin, and they exhibit CRM-like properties [80,81]. NA is converted to NAD+ via the Preiss–Handler pathway, while NAM and NR enter the NAD+ salvage pathway, playing a pivotal role in the maintenance of cellular NAD+ levels [82]. NMN is synthesized from NAM by nicotinamide phosphoribosyltransferase (NAMPT), the rate-limiting NAD+ biosynthetic enzyme in mammals, as well as from NR by nicotinamide riboside kinases (NRKs), effectively bypassing the need for NAMPT (Figure 5) [83]. Accumulating evidence suggests that NAD+ intermediates not only prolong healthspan and/or lifespan [84–86], compensating for reduced NAD+ levels during aging, but also seem to be an effective intervention for various age-associated diseases, including cardiovascular diseases [87,88], cancer [89,90], and neurodegenerative disorders [91].

NAD+ plays a dual and pivotal role in cellular responses, serving as an essential coenzyme for enzymes facilitating oxidation–reduction reactions and as a co-substrate for NAD+-consuming enzymes. These enzymes compete for bioavailable NAD+ and belong into three classes: the cyclic ADP-ribose (cADPR) synthases, such as CD38; the poly (ADP-ribose) polymerase (PARP) protein family; and the sirtuin family of deacetylases (Figure 5) [92,93].

Specifically, CD38 plays important roles in many physiological processes, including glucose homeostasis, inflammation, and neuroprotection. Its deletion and the subsequent elevation in NAD+ levels protect against high-fat diet (HFD)-induced obesity [94], inflammatory reactions of microglia and astrocytes, and ROS, while it improves CPZ-induced demyelination and neurodegeneration [95,96]. PARP proteins mediate ADP-ribosylation and act as DNA-damage sensors [97]. It has been reported that the upregulation of PARP

induces OL death, whereas its inhibition reduces CPZ-induced demyelination by suppressing p38 mitogen-activated protein kinases (p-38-MAPK) and JNK activation and increasing the activation of the PI3K/Akt pathway [98].

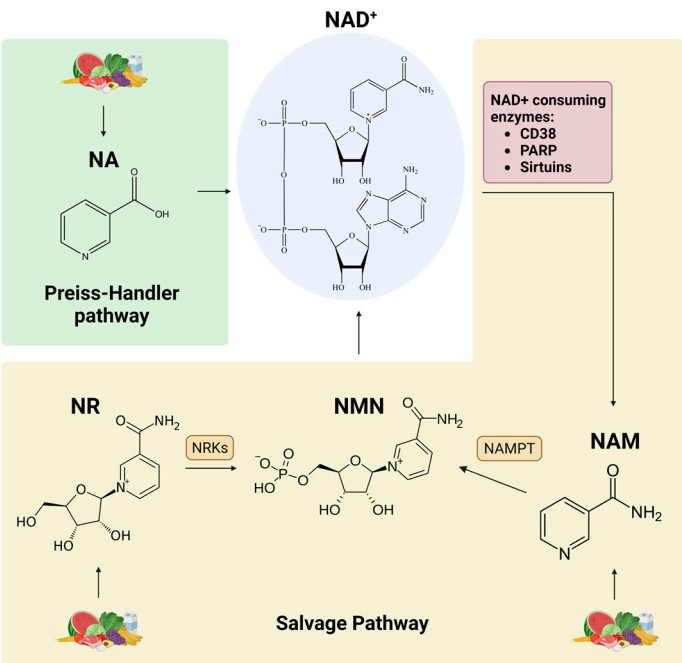

**Figure 5.** NAD+ biosynthetic pathways. NA is converted to NAD+ via the Preiss–Handler pathway, whereas NAM and NR enter the salvage pathway to produce NAD+. The NAD+ salvage pathway recycles NAM that is generated as a byproduct of the activity of NAD+-consuming enzymes: CD38, PARP, and sirtuins. NMN is synthesized from NAM by NAMPT as well as from NR by NRKs. PARP: poly (ADP-ribose) polymerase; NMN: nicotinamide mononucleotide; NAMPT: nicotinamide phosphoribosyltransferase; NRKs: nicotinamide riboside kinases. Created with BioRender.com (accessed on 21 November 2023).

Among the three classes of NAD+-consuming enzymes, sirtuins are the most well studied. Upon increased NAD+ levels, SIRT1 is activated, leading to the deacetylation of critical proteins of the autophagic pathway, including autophagy-related gene 5 (Atg5), Atg7, and microtubule-associated protein 1 light chain 3 (LC3), thus inducing autophagy, as well as to the deacetylation of transcription factors, like NF-κB, thus regulating inflammatory signaling [99,100]. Additionally, the administration of NA has been shown to inhibit vascular inflammation in vivo along with the suppression of the NOD-like receptor family pyrin domain-containing 3 (NLRP3) inflammasome in vascular endothelial cells in vitro via SIRT1 upregulation [101,102]. Apart from SIRT1, NAD+ supplementation has also been found to mediate the activation of sirtuin 2 (SIRT2). Specifically, NAD+ administration increased intracellular ATP levels via the activation of SIRT2, which regulates Akt phosphorylation in BV2 microglial cells [103], while NR treatment alleviated cisplatin-induced peripheral neuropathy in a SIRT2-dependent manner [104].

NAD+ supplementation was demonstrated to exert neuroprotective effects on various neurodegenerative diseases, including AD, PD, and HD. NR treatment reduced neuroinflammation in an APP/PS1 mouse model of AD, promoting the protective, phagocyting phenotype of microglia, while it also improved cognitive and synaptic functions [105]. A very recent study has highlighted NMN as a regulator of the gut microbiota, which exerted positive effects on AD [106]. Furthermore, NAM administration significantly protected against neuronal loss and attenuated motor dysfunction, oxidative stress, and neuroinflammation in a 1-methyl-4-phenyl-1,2,3,6-tetrahydropyridine (MPTP)-induced mouse model

of PD [107], while it also provided neuroprotection in the 3-nitropropionic acid-induced animal model of HD [108].

Apart from these pre-clinical studies, multiple clinical trials are currently being conducted to evaluate the safety and the effects of NAD+ precursors on neurological disorders. In particular, NR has been demonstrated to be orally bio-available without serious adverse effects [109,110]. A recent randomized, placebo-controlled, crossover trial of oral NR supplementation has indicated increased levels of NAD+ in plasma extracellular vesicles enriched for neuronal origin (NEVs). Increased NAD+ levels were accompanied by decreased levels of Aβ42 and of the activated kinases pJNK and pERK1/2, which are implicated in AD [111]. Furthermore, in a randomized, placebo-controlled, phase I clinical trial, oral NR administration increased cerebral NAD levels in individuals with PD, an effect that was associated with the downregulation of inflammatory cytokines and general clinical improvement [112]. Regarding NAM, a new clinical trial (NCT03061474) is investigating whether NAM can reduce the phosphorylation of the tau protein found in cerebrospinal fluid (CSF) in people with mild cognitive impairment or mild AD [113].

In the same context of neurodegenerative conditions, NAD+ precursors have been shown to exert beneficial effects on demyelinating, neuroinflammatory diseases, like MS. NAD+ treatment was shown to alleviate demyelination and neuroinflammation in both the spinal cord [114] and the optic nerve [115] of a murine EAE model. Its administration exhibited a marked reduction in the activation of microglia and astrocytes, as well as in the expression of pro-inflammatory cytokines, while it facilitated the expression of anti-inflammatory cytokines, thereby fostering a less inflammatory milieu in vivo, which could support the remyelination process (Figures 1 and 2) [114,115]. Notably, successful remyelination hinges upon the activities of OPCs. In response to myelin damage, these cells proliferate and migrate to lesion sites, where they mature to myelin-forming cells [116]. Guo et al. demonstrated that NAD+ supplementation effectively mitigates apoptosis among OLs and concurrently facilitates the recruitment and proliferation of OPCs in the optic nerve of mice with EAE. These beneficial effects of NAD+ in optic neuritis were orchestrated through the activation of the SIRT1-signaling pathway (Figure 3) [115]. Additionally, Wang et al. attributed the beneficial outcomes of NAD+ administration to the induction of autophagy because its inhibition abolished the protective effects of NAD+ [114]. Given the established role of SIRT1 deacetylase as an autophagy inducer [100], both studies converge on the critical role of the autophagic pathway in ameliorating EAE symptoms after NAD+ treatment.

In addition to NAD+ supplementation, numerous pre-clinical investigations have explored NAD+ precursors as potential therapeutic strategies for alleviating symptoms associated with MS. A recent study from our laboratory has revealed that NAM treatment resulted in a substantial augmentation in myelin production at the lesion site in the cc of an LPC-induced focal demyelination mouse model, concurrently with a reduction in microgliosis and astrogliosis (Figures 1 and 2). Importantly, NAM treatment did not exert a direct influence on oligodendrocyte lineage cells, thereby suggesting that it accelerated the overall myelin production under demyelinating conditions by mitigating both microgliosis and astrogliosis [117]. Furthermore, the same study indicated that NAM directly affected microglial and astrocyte polarization toward their anti-inflammatory phenotypes, thus fostering a beneficial, less inflammatory microenvironment for remyelination. In addition to NAM, NR pre-treatment attenuated inflammatory responses, glial activation, and subsequent neurodegeneration in the brain of a lipopolysaccharide (LPS)-injected mouse model (Figures 1 and 2) [118]. Despite these promising findings, the precise molecular mechanisms underpinning the NAD+-mediated regulation of glial activity remain elusive. Kaplanis et al. suggested that the shift in astrocytes toward their anti-inflammatory phenotype arises, at least in part, from the induction of autophagy, as observed in primary astrocyte cultures following NAM treatment (Figure 1) [117]. Notably, a recent study has identified a correlation between the induction of autophagy and the suppression of the inflammatory phenotype in astrocytes [119]. Regarding microglia, it appears that NAD+-dependent deacetylase SIRT2

inhibits pro-inflammatory responses through the deacetylation of NF-κB [120], indicating the participation of different NAD+-dependent pathways in glial phenotype commitment.

Among all seven sirtuins, SIRT2 is the most abundantly expressed in the brain, primarily residing in the cytoplasm of mature OLs but also present in neurons, astrocytes, and microglia [121,122]. Ma et al. further demonstrated that SIRT2 is predominantly expressed in the nuclei of postnatal OPCs during myelin development and changes its expression pattern in mature OLs, where it is found in the cytoplasm. Following a demyelination injury induced by LPC, SIRT2 is re-expressed in the majority of OPCs, primarily localizing within the OPC nuclear compartment in young adult mice. However, this re-expression and nuclear localization of SIRT2 declines with aging. Interestingly, β-NMN supplementation rescues SIRT2 nuclear localization in aged mice and affects the myelin status. In particular, it delays myelin aging under normal aging conditions and influences myelin compaction and thickness after a focal demyelinating LPC-induced lesion, thus enhancing remyelination by promoting the differentiation of OPCs (Figure 4) [123].

Enhanced remyelination was also observed in middle-aged animals upon niacin (NA) treatment in an LPC-induced focal demyelination mouse model [124]. Notably, the aging process is associated with delayed microglial recruitment to the lesion site and deficient phagocytosis, contributing to the establishment of an inhibitory microenvironment [125,126]. Rawji et al. demonstrated that NA administration enhanced the phagocytic activity of microglia in middle-aged animals, thus promoting the clearance of myelin debris from the lesion site and the recruitment of OPCs in favor of the remyelination process (Figure 4) [124]. Finally, NAM treatment promoted the maturation of OPCs and enhanced remyelination after stroke. NAM-treated animals had increased motor, sensory, and cognitive functions, and this functional remyelination was mediated by the BDNF/tropomyosin receptor kinase B (TrkB) pathway (Figure 3) [127]. Because BDNF is reported to enhance myelination via a direct effect on OLs [128], it is plausible that BDNF mediates the maturation of OPCs during remyelination.

These investigations highlight the favorable impacts of NAD+ and its precursors in age-related diseases, particularly in neurodegenerative and demyelinating diseases, like MS, rendering them promising therapeutic agents. Nevertheless, it is evident that distinct NAD+ precursors manifest their beneficial effects by targeting diverse molecules/pathways in various cell types of the CNS and within different mouse models of MS. Hence, it becomes imperative to discern the optimal precursor based on the considerations of absorption, kinetics, and specific MS symptoms. Finally, given that the majority of research efforts have centered on SIRT1 activation upon NAD+ treatment, it is equally important to delve into the roles of the other two classes of NAD+-depleting enzymes under demyelinating conditions.

## 4. Resveratrol

Resveratrol (3,4′,5-trihydroxystilbene) is a natural polyphenol that acts as a phytoalexin and is found in a wide variety of foods, including blueberries and peanuts, as well as grapes and products derived from them, like red wine [129,130]. Ever since resveratrol's potent anticancer properties were highlighted by Jang in 1997 [131], both experimental and epidemiological studies have been conducted to elucidate its diverse bioactivities and, consequentially, its health advantages. Interestingly, resveratrol has a positive impact on a wide spectrum of diseases, including heart diseases [132], diabetes [133], cancer [134], obesity [135], and neurodegenerative diseases [136,137], while it also exerts beneficial effects on aging [138,139].

Mechanistically, resveratrol is mainly associated with the activation of the NAD+-dependent deacetylase SIRT1. Once activated, SIRT1 can deacetylate the core proteins of the autophagic pathway, like Atg5 and Atg7, leading to the induction of this pathway [100,140]. Furthermore, the induction of SIRT1 activation by resveratrol necessitates its phosphorylation by LKB1 in multiple cell lines, subsequently resulting in the deacetylation of the peroxisome proliferator-activated receptor-gamma coactivator-1-alpha (PGC-1a) transcriptional co-activator, which regulates mitochondrial biogenesis and respiration [141,142].

Beyond SIRT1, resveratrol also engages AMPK as a target [143]. Its activation upon resveratrol treatment was found to rescue Aβ-mediated neurotoxicity in human neural stem cells (hNSCs) [144] as well as oxygen and glucose deprivation in human SH-SY5Y neuroblastoma cells [145]. Because AMPK has been shown to activate SIRT1 through an indirect increase in cellular NAD+ levels [146], there is clearly a dynamic interaction between the two pathways. Finally, it is worth mentioning that resveratrol exhibits anti-inflammatory properties by suppressing the production of ROS and downregulating NF-κB [147].

Resveratrol has been documented for its neuroprotective potential in various CNS disorders, notably AD and PD. In particular, resveratrol treatment has been shown to mitigate neuroinflammation and reduce Aβ accumulation in the brains of 3×Tg-AD mice [148]. Furthermore, it inhibited tau aggregation and cytotoxicity in vitro, and it reduced the levels of phosphorylated tau, neuroinflammation, and synapse loss in the brain of a PS19 mouse model of AD, thus rescuing the cognitive deficits [149]. Resveratrol treatment was also able to ameliorate motor and cognitive impairments in an A53T α-synuclein mouse model of PD by diminishing the levels of α-synuclein aggregates and reducing microgliosis, astrocytosis, and oxidative-stress levels within the brain [150].

Meanwhile, the effects of resveratrol on neurological disorders are evaluated through clinical trials. One of the first studies evaluating the effects of resveratrol on individuals with mild to moderate AD was conducted by Turner et al. in 2015. This randomized, double-blinded, placebo-controlled phase II study indicated that resveratrol is safe and well tolerated by patients, while it could also penetrate the blood–brain barrier because it was detectable in the CSF. However, neuroprotective benefits could not be detected in this study, while the longer AD duration, measured in years from the diagnosis, in the placebo-treated group should be taken into consideration [151]. In the next step, the same research group analyzed samples of CSF and plasma from a subset of AD subjects with CSF Aβ 42 concentrations of <600 ng/mL. In this subset analysis, resveratrol decreased the levels of metalloproteinase (MMP) 9 in the CSF, suggesting increased maintenance of the blood–brain barrier and reduced infiltration of immune cells, while it also regulated neuroinflammation, induced adaptive immunity, and mitigated progressive cognitive decline [152].

The established capacity of resveratrol to mitigate inflammation and attenuate gliosis in the context of neurodegenerative pathologies endows it with substantial therapeutic potential for addressing demyelinating diseases, like MS. MS is predominantly characterized by pronounced inflammatory responses orchestrated by microglia, which can participate in mechanisms of tissue repair and injury depending on their activation state [14,153]. Traditionally, microglia have been categorized into pro-inflammatory (M1) and anti-inflammatory (M2) phenotypes; however, this categorization appears to be simplistic. The use of new technologies, including single-cell RNA sequencing, has led to the identification of intermediate subpopulations that display a combination of pro- and anti-inflammatory markers, suggesting that microglial activation is a dynamic process [154–156]. A crucial requirement for the achievement of successful remyelination in MS is a switch toward the M2 activation state [157].

Resveratrol was shown to not only suppress microglial polarization toward the M1 phenotype but also promote the M2 phenotype of LPS-stimulated BV2 microglial cells in vitro and of microglial cells in vivo, in a model of systemic LPS administration that leads to brain inflammation (Figure 1) [158]. The neuroprotective role of resveratrol in microglial polarization was mainly attributed to PGC-1a activation, which can not only halt M1 polarization by suppressing NF-κB phosphorylation and the expression of inflammatory cytokines, like tumor necrosis factor alpha (TNF-α), but also interact with transcription factor signal transducer and activator of transcription 3 (STAT3) and STAT6, promoting the expression of the anti-inflammatory M2 markers arginase 1 (Arg1) and IL-10 [158]. A recent study has also indicated that resveratrol can promote the M2 microglial phenotype and reduce the degree of neuroinflammation after cerebral ischemia by inhibiting miR-155, a molecule linked to inflammatory processes and to the promotion of M1 polarization [159].

Interestingly, resveratrol-loaded macrophage exosomes, which addressed the low solubility of resveratrol, alleviated inflammation and symptom severity in EAE mice by targeting microglia [160]. These results indicate that resveratrol promotes the M2 microglial phenotype by mitigating inflammation, which is the main impediment of the remyelination process.

Like microglia, astrocytes can exert both detrimental and beneficial effects on remyelination depending on their neurotoxic (A1) or their neuroprotective (A2) phenotype, respectively [161]. Resveratrol treatment inhibited the expression of LPS-induced pro-inflammatory cytokines in both primary murine microglia and astrocytes, while it also reduced the expression of iNOS and the production of nitric oxide (NO) in these glial cell types (Figures 1 and 2) [162]. It is well established that in response to CNS demyelination, astrocytes become activated, proliferate, and form the glial scar, which impedes the remyelination process and is STAT3 dependent [163]. Resveratrol was found to attenuate reactive astrocyte proliferation and activation by downregulating STAT3 signaling in primary rat astrocyte cultures (Figure 2) [164]. These in vitro results suggest that resveratrol could be a promising agent for facilitating remyelination in vivo by regulating the glial scar and establishing a less inflammatory microenvironment.

Reduced inflammation following the administration of resveratrol was further observed in a mouse model of CPZ-induced demyelination. In this context, resveratrol also reduced lipid peroxidation and countered the negative impact of CPZ on the mitochondrial respiratory chain, as assayed by increased cytochrome oxidase activity and ATP levels [165]. Recent studies have established a connection between oxidative stress and the compromised differentiation capacity of OLs, consequently contributing to the process of demyelination [166]. Thus, resveratrol's effect on the alleviation of oxidative stress could be correlated with the myelin status recovery. Indeed, Ghaiad et al. showed that resveratrol increased myelin basic protein (MBP) expression levels and the stain intensity of Luxol fast blue (LFB), while it also improved balance and motor coordination that were impaired in CPZ-intoxicated mice. These biochemical, histological, and behavioral results indicate that resveratrol reversed CPZ-induced demyelination and enhanced the remyelination process (Figure 3) [165].

Similar effects of resveratrol on balance and motor coordination as well as on enhanced myelin repair in CPZ-treated mice were reported by Samy et al. in 2023. However, the significant improvement in behavioral tests was incomplete compared to control animals, whereas despite the increased number of myelinated axons in the cc, not all the repaired myelin was compacted, and resveratrol failed to upregulate MBP expression levels. These conflicting effects of resveratrol were attributed to different disease-induction and treatment protocols [167]. Interestingly, Samy et al. correlated the positive effects of resveratrol with the induction of autophagy, which is the main result of CR and was interrupted at a late stage in CPZ-treated mice. The induction of the autophagic flux and the successful autophagic degradation upon resveratrol administration involved the activation of the SIRT1/forkhead box protein O1 (FOXO1) pathway [167]. However, the cell-autonomous effect of resveratrol regarding the induction of autophagy and its beneficial effects on remyelination are yet to be determined.

Previous studies have highlighted the advantageous impact of resveratrol on myelination within the PNS. Using an in vitro system comprising a dorsal root ganglion (DRG)/SC co-culture, researchers discerned that resveratrol enhanced myelination, an effect that was mediated, at least in part, by SIRT1 activation in SCs, which serve as the myelinating cells of the PNS [168]. Furthermore, resveratrol induced autophagy in SCs, leading to myelin sheath degeneration in the early stages of nerve injury and, thus, promoting recovery from sciatic nerve crush injury [169]. It is important to acknowledge that myelin clearance represents a critical phase in the regeneration process following peripheral nerve injury [170].

In the context of the CNS, resveratrol mediates protective effects on OLs by preventing LPS-mediated cytotoxicity and reducing the abundance of ROS [171], while it also promotes the survival, migration, proliferation, and differentiation of OPCs in a rat model of

ischemic cerebral injury (Figure 3) [172]. Furthermore, resveratrol has demonstrated neuroprotective effects on a chronic EAE mouse model because its administration attenuated the neuronal loss of retinal ganglion cells (RGCs) [173]. Correspondingly, recent research has corroborated resveratrol's neuroprotective effects, which were attributed to its capacity to promote autophagic activity in a mouse spinal cord injury model [174]. Although the existing evidence pertaining to the roles of resveratrol in myelination and regeneration in the CNS is limited, it is plausible that these processes are regulated, similarly to PNS, by the induction of the autophagic pathway in OLs, which are the myelinating cells of the CNS.

Recent research has emphasized the significance of oligodendroglial autophagy in OL maturation and the maintenance of CNS myelin [21,22]. The activation of the autophagy inducer SIRT1 has been shown to mediate the proliferation and regeneration of OPCs in the white matter of neonatal mice under hypoxic conditions [175]. During adulthood, on the other hand, the genetic ablation of SIRT1 increased the pool of OPCs after focal demyelination, promoting the remyelination process and, thus, indicating the temporally restricted role of SIRT1 in glial regeneration following brain injury [176]. Furthermore, recent findings have suggested that SIRT1 is upregulated in OPCs in EAE and likely plays a role in remyelination [177]. Given that resveratrol is predominantly associated with SIRT1 activation, these pieces of evidence underscore the importance for investigating the cell-autonomous effects of resveratrol on OLs under demyelinating conditions and its potent role as a therapeutic agent for MS.

## 5. Conclusions

CRMs have gained significant attention within the scientific community, emerging as promising agents capable of emulating numerous effects that are typically induced by CR. Notably, many CRM candidates can induce autophagy, prolong lifespan and/or healthspan, and mitigate the symptoms of age-related diseases, all without the subjective discomfort associated with CR. In addition, CRMs have been shown to exert beneficial effects on demyelinating neuroinflammatory diseases, like MS, by modulating the profile of glial cells, ultimately facilitating the remyelination process. In particular, apart from targeting the migration, proliferation, and differentiation of OLs, CRMs affect microglia and astrocytes by promoting their protective phenotypes, thereby establishing a less inflammatory microenvironment that supports remyelination.

This translational research on CRMs has now progressed to the clinical phase because there is an unmet need to verify their favorable effects through clinical trials. Presently, there are ongoing clinical trials investigating the effects of metformin on endogenous neural progenitor cells in children or young adults with MS (NCT04121468) [178], as well as the safety of metformin for the treatment of progressive MS (NCT05349474) [179]. There is also a clinical trial (NCT05740722), currently recruiting patients, that aims to evaluate the safety and efficacy of NR in the treatment of patients with progressive MS [180]. Despite the large number of in vitro and in vivo studies using animal models of MS, clinical evidence for the protective role of polyphenols in MS patients is restricted, encompassing only a few compounds, like curcumin [181]. However, taking into consideration the beneficial effects of resveratrol on mouse models of MS, as well as its established safety and its ability to modulate neuroinflammation in patients with AD, clinical trials need to be conducted to evaluate the potential of resveratrol to mitigate the symptoms of patients with MS. It is important, though, that aspects such as bioavailability, cellular uptake, systemic distribution, and organ-specific effects are settled. A recent study has used resveratrol nanoparticles to address the poor water solubility and bioavailability of resveratrol in an EAE mouse model. The results have suggested that the nanoparticles increased the bioavailability of the resveratrol and exerted neuroprotective effects by reducing the loss of retinal ganglion cells [182]. Moreover, it is possible that the combination of mechanistically different CRMs will have synergistic effects, thereby maximizing their positive impact.

Additional research also needs to be undertaken to elucidate the influences of distinct CRMs on the myelin sheaths of the elderly. As the human brain ages, the capacity of OPCs

to differentiate in mature myelinating OLs significantly declines [183,184]. Notably, the transition from relapsing–remitting to progressive MS takes place at around the same age in MS patients, indicating that it is mostly age rather that disease-duration dependent [185]. Apart from aging, both the disease duration and anatomic sites of lesions affect the remyelination potential of MS patients. It is suggested that remyelination is a more frequent event at the beginning of the disease than in the chronic phase, when remyelination is scarce and predominantly confined to the peripheries of lesions [186]. Regarding the location, cortical lesions are more extensively remyelinated than white-matter ones [187]. Thus, it becomes evident that even though remyelination is highly efficient in animal models and is commonly observed in MS patients, it varies considerably between lesions and between individuals, and these facts should be taken into account during the design of clinical trials. Finally, neuropathological studies reveal that some lesions of MS patients lack OPCs [188], whereas other lesions are characterized by a great number of OPCs with an impaired differentiation capacity [189], indicating that proliferating or differentiating agents should be used according to the lesion status. Consequently, even though spontaneous remyelination exists in humans following a demyelination insult, many obstacles need to be overcome not only for the development of efficient MS treatments based on CRMs but also for the proper evaluation of remyelination. However, despite the imperative for validating CRMs as therapeutic approaches, the existing body of evidence corroborates the considerable potential of these autophagy inducers against MS and multiple age-related diseases.

**Author Contributions:** D.K. (Despoina Kaffe), writing—original draft preparation and visualization; S.I.K., writing—review and editing; D.K. (Domna Karagogeos), conceptualization, supervision and editing; D.K. (Despoina Kaffe), S.I.K. and D.K. (Domna Karagogeos), funding acquisition. All authors have read and agreed to the published version of the manuscript.

**Funding:** This research was funded by the Hellenic Foundation for Research and Innovation (HFRI) under grant agreement No. 1676, the Hellenic Academy of Neuroimmunology (HELANI), and the Fondation Sante (The Sidney Altman Scholarship Program). De.K was supported by an Onassis Foundation Scholarship (G ZS 004-1/2022-2023).

**Institutional Review Board Statement:** Not applicable.

**Informed Consent Statement:** Not applicable.

**Data Availability Statement:** Not applicable.

**Acknowledgments:** The authors would like to thank all the members of the Karagogeos laboratory for helpful and thought-provoking conversations.

**Conflicts of Interest:** The authors declare no conflict of interest.

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
