# Peer review of "The Roles of Caloric Restriction Mimetics in Central Nervous System Demyelination and Remyelination"

_cimb, doi:10.3390/cimb45120596_

Round 1

Reviewer 1 Report

Comments and Suggestions for Authors

This is an exceptionally well written review that outlines the mechanistic drivers of remyelination or quite possibly myelin maintenance in neurological conditions that may well propagate progression that include but are not limited to multiple sclerosis in the chronic demyelinating pathological setting. The authors very eloquently outline calorie restriction as a driver of myelinogenesis through autophagy. They specify details of Metformin, NAD+ precursors and resveratrol as excellent enhancers of metabolic CRM-like properties. There are some key attributes about the manuscript that I would like resolved prior to publication.

1.      There is a lot of discussion related to neurorepair ascribed to the process of remyelination through the engagement, proliferation and maturation from OPCs. Although experimentally this can be observed it is by no means achievable in humans to a robust degree outside of possibly the optic nerves primarily due to the limitation of old plaques when we consider Secondary progressive MS lesions with profound gliotic scares. This needs to be made very clear in the manuscript for accuracy.

2.      Again there is no discussion for the best possible outcome in people living with MS, that would include the neuroprotective mechanisms governing the agents currently under investigation, promoting mature oligodendrocyte preservation and potential remyelination from these cells on their nearby denuded axons.

3.      There was discussion of Metformin and its inactivation of mTORC1 leading to remyelination outcomes on pg3 and again on pg 5 and may well be counterintuitive with the very robust data generated from Wendy Macklin’s group that clearly define the activation of mTORC1 as the main driver of remyelination in Cuprizone models – See: Bercury et al 2014 J Neurosci; Ishii et al., 2019; Sawade et al., 2020 Nat Comm etc.

4.      Please rephrase the M2 phenotype defined on p11 as this is very much an older nomenclature and requires redefinition.

Author Response

Responses to Reviewer 1

Comment 1

There is a lot of discussion related to neurorepair ascribed to the process of remyelination through the engagement, proliferation and maturation from OPCs. Although experimentally this can be observed it is by no means achievable in humans to a robust degree outside of possibly the optic nerves primarily due to the limitation of old plaques when we consider Secondary progressive MS lesions with profound gliotic scares. This needs to be made very clear in the manuscript for accuracy.

Response

We have included a paragraph in the conclusions section which we believe is now more clear (lines 677-697)

Comment 2

Again there is no discussion for the best possible outcome in people living with MS, that would include the neuroprotective mechanisms governing the agents currently under investigation, promoting mature oligodendrocyte preservation and potential remyelination from these cells on their nearby denuded axons.

Response

We have looked for clinical trials for CRMs and MS but could not find published results (ongoing). For this reason, we have included information on clinical trials for CRMs and other neurodegenerative diseases (lines 165-176, 404-416, 530-543) while in the conclusions we have included the ongoing clinical trials with CRMs and MS (with metformin, NR, polyphenols, lines 655-663). No information on a clinical trial with resveratrol in MS was found in the literature.

Comment 3

There was discussion of Metformin and its inactivation of mTORC1 leading to remyelination outcomes on pg3 and again on pg 5 and may well be counterintuitive with the very robust data generated from Wendy Macklin’s group that clearly define the activation of mTORC1 as the main driver of remyelination in Cuprizone models

Response

In the paragraph that starts in line 240 and ends in line 256 we review the relationship of mTOR and myelination and our concluding phrase is that “more research is warranted to elucidate the interactions between AMPK and mTOR upon metformin administration”

Comment 4

Please rephrase the M2 phenotype defined on p11 as this is very much an older nomenclature and requires redefinition.

Response

It has been rephrased (page 12-13). 

Reviewer 2 Report

Comments and Suggestions for Authors

In this review Kaffe and collegues wish to discuss how caloric restriction might affect demyelination remyelination. This is an interesting topic, very dispersed in the literature, so a review is welcome.

The difficulty comes when after a general introduction easy to read they  say that they will focus on substances “ capable of modulating the autophagic machinery within myelinating glial cells of the CNS”

Following this the text put in connection CR with remyelination but the cause of the recovery remains unclear and could be related to several factors. Indeed, they state that “Nevertheless, due to the systemic and extensive impact of CR, unravelling the specific signalling pathways and the exact mechanisms underpinning its favourable effects mediated via autophagy can prove to be a complex endeavor. “

So the reader remains with the question if really there is a connection , which are the data in favour and which against. Unfortunately, they do not specify how autophagic machinery modulates remyelination in normal condition nor they provide the evidence that link selected compounds to autophagy.

This makes the reader to lose the line of the review, making difficult to understand the relationship among the statet aim and the following arguments

Although the different aspect of each drug action in different pathways and cell type are well described, In general I find difficult to follow the connection between one argument to the other. Maybe they should simplify sentences as in general I find difficult to follow the line that bring to their conclusion.

Minor point : From line 361 to line 371 there is a self-citation.

This is a relevant paper to cite in this contest Shamsher E, Khan RS, Davis BM, Dine K, Luong V, Somavarapu S, Cordeiro MF, Shindler KS. Nanoparticles Enhance Solubility and Neuroprotective Effects of Resveratrol in Demyelinating Disease. Neurotherapeutics. 2023 Jul;20(4):1138-1153. doi: 10.1007/s13311-023-01378-0. Epub 2023 May 9. PMID: 37160530; PMCID: PMC10457259.

Author Response

Comment 1

Τhey do not specify how autophagic machinery modulates remyelination in normal condition

Response

We have now expanded on this information in the introduction which we hope the reviewer finds satisfactory (lines 63-86)

Comment 2

Nor they provide the evidence that link selected compounds to autophagy.

Response

We have highlighted in the text with yellow highlights the parts where we mention that the compounds induce autophagy (pg2 lines 96-98; pg3 lines 122-125; 140-143; 150-151; pg5 lines 229-230; pg6 lines 236-239; pg7 lines 302-304; pg7 lines 314-316; pg9 lines 381-384; pg10 lines 429-435; pg11 lines 451-455; pg 12 lines 506-508; pg 13-14 lines 604-610; pg14 lines 612-617; pg14 lines 624-633; pg14 lines 636-638).

Comment 3

Maybe they should simplify sentences as in general I find difficult to follow the line that bring to their conclusion.

Response

Sentences were restructured as much as possible to make the points more clear

Minor point

-self citation:

Response

Indeed, it is a self citation and it should be cited as it is the only work on NAM that describes these results which we consider significant.

-paper Shamsher et al., 2023: 

Response

This suggested reference was included in the conclusions section

Round 2

Reviewer 1 Report

Comments and Suggestions for Authors

The Authors have now edited the manuscript appropriately for publication.